# Food Protein-Induced Enterocolitis Syndrome in Children with Down Syndrome: A Pilot Case-Control Study

**DOI:** 10.3390/nu14020388

**Published:** 2022-01-17

**Authors:** Fumiko Okazaki, Hiroyuki Wakiguchi, Yuno Korenaga, Kazumasa Takahashi, Hiroki Yasudo, Ken Fukuda, Mototsugu Shimokawa, Shunji Hasegawa

**Affiliations:** 1Department of Pediatrics, Yamaguchi University Graduate School of Medicine, 1-1-1 Minamikogushi, Ube, Yamaguchi 755-8505, Japan; fumiko@yamaguchi-u.ac.jp (F.O.); kyuno@yamaguchi-u.ac.jp (Y.K.); p-sama@yamaguchi-u.ac.jp (K.T.); yasudo@yamaguchi-u.ac.jp (H.Y.); kfu10000@yamaguchi-u.ac.jp (K.F.); shunji@yamaguchi-u.ac.jp (S.H.); 2Department of Biostatistics, Yamaguchi University Graduate School of Medicine, 1-1-1 Minamikogushi, Ube, Yamaguchi 755-8505, Japan; moto@yamaguchi-u.ac.jp

**Keywords:** cow’s milk allergy, food allergy, food hypersensitivity, gastrointestinal disorder, non-IgE-mediated food hypersensitivity disorder, wheat allergy

## Abstract

Food protein-induced enterocolitis syndrome (FPIES) is a non-immunoglobin E-mediated food hypersensitivity disorder. However, little is known about the clinical features of FPIES in patients with Down syndrome (DS). Medical records of children with DS diagnosed at our hospital between 2000 and 2019 were retrospectively reviewed. Among the 43 children with DS, five (11.6%) were diagnosed with FPIES; all cases were severe. In the FPIES group, the median age at onset and tolerance was 84 days and 37.5 months, respectively. Causative foods were cow’s milk formula and wheat. The surgical history of colostomy was significantly higher in the FPIES group than in the non-FPIES group. A colostomy was performed in two children in the FPIES group, both of whom had the most severe symptoms of FPIES, including severe dehydration and metabolic acidosis. The surgical history of colostomy and postoperative nutrition of formula milk feeding may have led to the onset of FPIES. Therefore, an amino acid-based formula should be considered for children who undergo gastrointestinal surgeries, especially colostomy in neonates or early infants. When an acute gastrointestinal disease is suspected in children with DS, FPIES should be considered. This may prevent unnecessary tests and invasive treatments.

## 1. Introduction

Food protein-induced enterocolitis syndrome (FPIES) is a non-immunoglobin E (IgE)-mediated food hypersensitivity disorder that primarily affects formula-fed infants and young children [1,2]. The clinical manifestation of FPIES is characterized by profuse and repetitive vomiting, usually occurring within a few hours of feeding, accompanied by lethargy and pallor; diarrhea may also occur within 24 h. Symptoms usually resolve hours after the elimination of the causative food from the diet. Infants who consume foods such as cow’s milk or soy-based formula daily may experience chronic weight loss and failure to thrive [3]. FPIES is considered part of a spectrum of allergic diseases that affect only the gut [4]. Although the true incidence of FPIES is not known, large population-based cohort studies from Israel and Spain have reported the cumulative incidence of cow’s milk FPIES to be 0.34% and 0.35%, respectively [5,6]. Furthermore, the lifetime prevalence of physician-diagnosed FPIES was reported in the United States, with an estimated prevalence of 0.51% [7]. Because FPIES can be diagnosed clinically and an intestinal biopsy is not performed routinely, little is known about this condition. Specifically, the pathophysiology of FPIES has not been clearly defined and requires further characterization [4].

Several immunological alterations have been reported to be associated with FPIES. Previous studies have suggested the involvement of antigen-specific T cells and their production of proinflammatory cytokines that regulate the permeability of the intestinal barrier [8]. A recent study showed that the levels of transforming growth factor (TGF)-β and interleukin (IL)-10 were significantly lower in children with FPIES than in those with tolerance acquisition [4,6,8]; therefore, TGF-β and IL-10 were proposed as potential biomarkers of FPIES [4,9].

Down syndrome (DS) is caused by a trisomy of human chromosome 21 and occurs in approximately one in 1000 newborns [10,11,12]. Because the immune system in individuals with DS is altered, accompanied by signs of deficiency and dysregulation, there is a high incidence and prevalence of autoimmune diseases among such individuals [13]. A recent cohort study revealed a lower percentage of allergic sensitization in children with DS than in healthy controls, and no DS children aged 0–2 years had allergic sensitization [14].

We have previously reported two DS children with FPIES [15]. The clinical course of FPIES suggested that this condition may be more severe and require a longer duration to establish tolerance in DS children than in those without DS. However, little is known about the clinical features of FPIES in patients with DS. This study aimed to clarify the clinical features of FPIES in children with DS.

## 2. Materials and Methods

### 2.1. Study Population and Data

This was a single-center, retrospective study which was approved by the Institutional Review Board of Yamaguchi University Hospital (H2020-198). Informed consent was obtained from the parents of each patient before their inclusion in the study, and data were collected from patient medical records at our hospital.

Between 1 January 2000 and 31 December 2019, all 62 children with DS born at our hospital or referred to our hospital during the neonatal period were enrolled in this study (Figure 1). Nineteen of the 62 children with DS were excluded from the analysis because they were lost to follow-up. Three children had insufficient data, seven moved shortly after birth, seven were transferred to another hospital for cardiac operations, and two died within the first year of life. Finally, the clinical course of 43 children with DS was followed up for more than 1 year. The DS patients were diagnosed according to chromosomal examinations. FPIES patients were diagnosed according to the criteria of the International Consensus Guidelines for the Diagnosis and Management of FPIES [16].

### 2.2. Outcome Measurements

Outcome measurements included demographic characteristics of the FPIES and non-FPIES groups in children with DS and clinical features of FPIES in children with DS. Demographic characteristics included sex, gestational age, birth weight, delivery type, neonatal asphyxia, neonatal jaundice, nutrition, comorbidities, surgical history, serum total IgE level, and antigen-specific IgE level in those below 12 months of age. Serum total IgE level was detected by IgE-LATEX “SEIKEN” (Denka Seiken, Tokyo, Japan), and antigen-specific IgE level was detected by ImmunoCAP (Thermo Fisher Scientific, Uppsala, Sweden). Clinical features of FPIES included age at onset, diagnosis, tolerance, causative foods, clinical symptoms, and severity.

### 2.3. Statistical Analyses

Demographic characteristics, surgical history, and serum total IgE were summarized using descriptive statistics or contingency tables. The Fisher’s exact test or the Mann–Whitney U test was used to compare variables between two groups. Statistical significance was set at *p* < 0.05. Statistical analyses were performed using JMP Pro version 14 (SAS Institute, Cary, NC, USA).

## 3. Results

Among 43 children with DS, five (11.6%) were diagnosed with FPIES (Figure 1). In the FPIES and non-FPIES groups, sex, gestational age, and median birth weight were approximately the same (Table 1). In the FPIES and non-FPIES groups, more than 60% of children were born by vaginal delivery. Neonatal asphyxia was observed in less than one-fourth, and neonatal jaundice in approximately half of the children. Neonatal asphyxia in this study was defined as an Apgar score ≤7 at one min after birth, and neonatal jaundice was defined as requiring phototherapy. There were no breast milk-only infants in either group, and mixed feeding infants accounted for approximately 80% of the total children in both groups.

Comorbidities of cardiac disease, gastrointestinal disease, and hematological disorder were found in most children with DS in both groups (Table 1). There was no significant difference in comorbidities between the two groups; however, gastrointestinal disease was more common in the FPIES group (40.0% vs. 10.5%, respectively, *p* = 0.136). There was no significant difference in the total surgical history of children with DS between the two groups; however, surgery for gastrointestinal disease was more common in the FPIES group (Table 2). Furthermore, the surgical history of colostomy was significantly higher in the FPIES group than in the non-FPIES group (40.0% vs. 2.6%, respectively, *p* = 0.032).

The serum total IgE levels were determined in all children less than 12 months of age in the FPIES group and six of 38 children in the non-FPIES group (Table 3). The median serum total IgE levels were less than the detection limit (<11 IU/mL) in both groups, and there was no significant difference between the groups. The antigen-specific IgE levels were less than the detection limit (<0.35 kU_A_/L) in all children in the FPIES group (Table 4), while they were not determined for the non-FPIES group.

In the FPIES group, the median age of onset in five cases was 84 days (Table 4). The causative foods were cow’s milk formula in four cases and wheat in one case. Repetitive vomiting and diarrhea were observed in all five cases. Bloody stools and abdominal distension were observed in two cases. Fever and metabolic acidosis were observed in four cases. Severe dehydration was observed in two cases with metabolic acidosis. In the case of severe dehydration, the causative food was cow’s milk formula. All five cases were diagnosed as severe according to the guidelines [16], and the patients required infusion.

Cardiac disease was observed in three cases. In Case 2, pulmonary artery banding was performed 50 days after birth for pulmonary hypertension with tetralogy of Fallot. Thirty-four days after the surgery, the brand of formula milk was changed, and vomiting and bloody stool appeared. Intracardiac repair was performed at the age of 12 months for tetralogy of Fallot. In Case 4, the patient did not require surgery for atrial septal defect. In Case 5, intracardiac repair was performed at the age of 19 months for atrioventricular septal defect. Gastrointestinal disease was observed in two cases. In Case 4, colostomy was performed 63 days after birth for rectovaginal fistula. In Case 5, colostomy was performed 1 day after birth for imperforate anus. The postoperative nutrition was formula milk for both cases. In Cases 4 and 5, the closure of colostomy was performed at the age of 23 months and 16 months, and tolerance was acquired at 49 months and 18 months of age, respectively. In Case 3, there were multiple episodes of repetitive vomiting after ingestion of wheat food, such as “udon” noodles and pancakes, at 10 months of age. Case 3 was a recent case in which the parent did not agree to the second oral food challenge; therefore, we could not confirm the acquisition of tolerance. In Case 1, the oral food challenge was performed six times, and finally, it took 11 years for the subject to acquire tolerance to cow’s milk formula. The median age of tolerance in the four cases was 37.5 months.

## 4. Discussions

This report describes the clinical features of FPIES in children with DS. In this study, no significant differences were seen in the total surgical history between the two groups; however, surgery for gastrointestinal disease was more common in the FPIES group. Furthermore, the surgical history of colostomy was significantly higher in the FPIES group than in the non-FPIES group (Table 2). In Cases 4 and 5 (Table 4), surgery for colostomy and postoperative nutrition of formula milk feeding may have led to the onset of FPIES. In Case 4, shortly after surgery, the nutrition of formula milk caused serious symptoms, including severe dehydration and metabolic acidosis. However, in Case 5, formula feeding was resumed after surgery, and repetitive vomiting was observed after 4 months. Subsequently, 7 months after the surgery, watery diarrhea appeared following the administration of antibacterial agents after cardiac catheterization, which caused shock. Both cases were the most severe in this study, and aggressive intervention was required. Finally, the stool form was normalized by the administration of an amino-based formula. In neonates and infants, formula milk after surgery was a risk factor of non-IgE-mediated gastrointestinal food allergies when compared to breast milk [17]. Therefore, an amino acid-based formula should be considered for children who undergo gastrointestinal surgeries, especially colostomy in neonates or early infants. In our study, there were no breast milk-only infants in either group, and mixed feeding infants accounted for approximately 80% of the total children in both groups. The mothers had been instructed that they could breastfeed or formula feed, but a retrospective review of the medical records showed that no children in either group were fully breastfed. We believe that a prospective study is needed to determine whether the active recommendation of full breastfeeding could reduce the incidence of FPIES. In addition, the median serum total IgE was less than the detection limit (<11 IU/mL) in both groups (Table 3). The serum IgE sensitization in infants with DS was low, which was the same as previously reported [14]. This suggests that non-IgE-mediated food hypersensitivity disorder is more likely to occur in DS during infancy.

In this study, all the cases were severe (Table 4). Repetitive vomiting and diarrhea were observed in all cases. Metabolic acidosis and severe dehydration were observed in four patients whose causative food was cow’s milk formula. This suggests that cow’s milk allergy may cause more serious symptoms than wheat allergy in FPIES patients with DS. There have been no reports of wheat-induced FPIES in patients with DS, besides Case 3 (Table 4). In Case 1 (Table 4), the patient took more than 10 years to acquire tolerance to cow’s milk. Immune disorders in DS may be associated with high incidence, severity, and difficulty in the acquisition of tolerance.

The CD4+ CD25+ Foxp3+ regulatory T (Treg) cells, which account for almost 10% of peripheral CD4+ T cells, are essential for the balance between pro- and anti-inflammatory responses at mucosal surfaces. There are two subsets of Treg cells, natural Treg (nTreg) cells and induced Treg (iTreg) cells [18]. While nTreg cells are generated in the thymus, iTreg cells arise from peripheral naïve T cells [18]. The thymus in patients with DS presents profound anatomical and architectural abnormalities [19], which may cause alterations in the maturation process of nTreg cells [20]. Individuals with DS manifest with an over-expressed peripheral nTreg population with a defective inhibitory activity that may partially explain the increased frequency of autoimmune diseases [21]. In a recent study, a higher proportion of circulating nTreg cells specific for cow’s milk protein was revealed in infants who had outgrown cow’s milk FPIES, suggesting that mucosal induction of tolerance against dietary antigens was associated with the development of nTreg cells [20]. Under steady-state conditions, TGF-β and IL-10 maintain peripheral and gut tolerance [22]. Children with active FPIES against cow’s milk have deficient T cell-mediated TGF-β response to casein; therefore, TGF-β could be a promising biomarker for identifying children who are likely to experience FPIES reactions to this allergen [9]. These results suggest that the suppressive action of cow’s milk-specific nTreg contributes to the production of TGF-β in children with resolved FPIES to cow’s milk [8]. TGF-β is a pleiotropic cytokine that is best known for its regulatory activity and induction of peripheral tolerance [22]. Unlike most other cytokines, TGF-β is produced by many immune and non-immune cells, and virtually, all cell types are responsive to this pleiotropic cytokine [23]. Similarly, IL-10 levels are significantly lower in patients with cow’s milk FPIES; however, the levels tend to increase in children with resolved FPIES to cow’s milk [4]. IL-10 is a key regulator of the immune system that acts by limiting the inflammatory response, which could otherwise cause tissue damage and is essential for the homeostasis of the immune system, especially in the gastrointestinal tract [22]. Therefore, increased IL-10 expression is also associated with tolerance acquisition in patients with FPIES [24]. IL-10 is produced mainly by T helper 2 cells, T helper 1 cells, nTreg cells, and natural killer T cells during chronic antigen stimulation [25]. In the absence of effector cytokines and in the presence of high concentrations of TGF-β, naïve CD4+ T cells are converted into iTreg cells that produce TGF-β and IL-10. Considering this information, individuals with DS, who have deficient nTreg inhibitory activity and reduced inhibitory activity of effector cytokines, may be more likely to develop FPIES and with more severity; thus, patients may take a longer time to acquire tolerance. In addition, humoral responses have been investigated in FPIES. Lower levels of antigen-specific IgE, IgA, and IgG4 have been found in patients with FPIES compared with those in patients with resolved FPIES [4]. Therefore, humoral immune responses may also be involved in the pathophysiology of FPIES.

FPIES in adults has been reported for many years; however, only recently, adult case series have been published in the peer-reviewed literature [26,27,28,29,30]. The dramatic symptoms of acute FPIES are usually triggered by shellfish and fish, whereas more chronic gastrointestinal symptoms have been attributed to cow’s milk, wheat/gluten, and eggs. The predominance of female adult patients with FPIES (88%) is striking [31] and has been also reported in some reports [27,28,29]. Contrastingly, infantile FPIES is slightly more common in males [31]. Also, there are no reports of FPIES in adult patients with DS. Further large multicenter studies are needed to better characterize adult FPIES.

The limitations of our study include single-center experience, small sample size, and limited follow-up period. Furthermore, due to the retrospective nature of the study, mild cases of this condition may have been overlooked. In addition, detailed cytokine profiles were not sufficiently examined in this study. Larger prospective multinational cohort studies are required to better understand the true incidence, risk factors, and clinical features of FPIES in patients with DS.

## 5. Conclusions

In our study, five (11.6%) of the 43 children with DS were diagnosed with FPIES, and all cases were severe. The surgical history of colostomy and postoperative nutrition of formula milk feeding may have led to the onset of FPIES; furthermore, cases involving colostomy were the most severe ones in our study. Therefore, an amino acid-based formula should be considered for children who undergo gastrointestinal surgeries, especially colostomy in neonates or early infants. Serum IgE sensitization in infants with DS was low, as previously reported; thus, non-IgE-mediated food hypersensitivity disorder is more likely to occur in DS during infancy. When an acute gastrointestinal disease is suspected in children with DS, FPIES should be considered. This may prevent performing unnecessary tests and invasive treatments.

## Figures and Tables

**Figure 1 nutrients-14-00388-f001:**
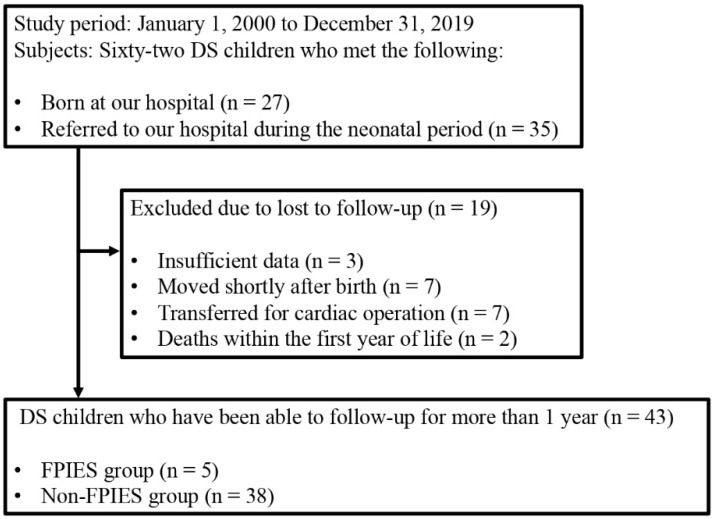
Flow diagram of the subject selection process. DS, Down syndrome; FPIES, food protein-induced enterocolitis syndrome.

**Table 1 nutrients-14-00388-t001:** Demographic characteristics of children with DS.

	FPIES (*n* = 5)	Non–FPIES (*n* = 38)	*p*-Value
**Male (n (%))**	3 (60.0)	20 (52.6)	1.000 †
**Median gestational age (weeks (range))**	37 (31–38)	38 (30–40)	0.168 ‡
**Median birth weight (g (range))**	2803 (1714–3224)	2802 (978–3500)	0.910 ‡
**Delivery type (n (%))**			
Vaginal	3 (60.0)	27 (71.1)	0.630 †
Emergency cesarean section	1 (20.0)	8 (21.1)	1.000 †
Elective cesarean section	1 (20.0)	3 (7.9)	0.402 †
**Neonatal asphyxia (n (%))**	1/4 (25.0)	5/37 (13.5)	0.483 †
**Neonatal jaundice (n (%))**	2/4 (50.0)	16/37 (43.2)	1.000 †
**Nutrition (n (%))**			
Mixed feeding	4 (80.0)	28/32 (87.5)	0.538 †
Formula milk	1 (20.0)	4/32 (12.5)	0.538 †
Breast milk only	0 (0)	0/32 (0)	1.000 †
**Comorbidities (n (%))**			
**Cardiac disease**	3 (60.0)	26 (68.4)	1.000 †
ASD	1 (20.0)	7 (18.4)	1.000 †
TOF/DORV	1 (20.0)	5 (13.2)	0.547 †
PDA	0 (0)	4 (10.5)	1.000 †
AVSD	1 (20.0)	3 (7.9)	0.402 †
VSD + PDA	0 (0)	3 (7.9)	1.000 †
VSD	0 (0)	1 (2.6)	1.000 †
VSD + ASD	0 (0)	1 (2.6)	1.000 †
VSD + ASD + PDA	0 (0)	1 (2.6)	1.000 †
PVP	0 (0)	1 (2.6)	1.000 †
**Gastrointestinal disease**	2 (40.0)	4 (10.5)	0.136 †
Duodenal atresia	0 (0)	3 (7.9)	1.000 †
Imperforate anus	1 (20.0)	1 (2.6)	0.222 †
Rectovaginal fistula	1 (20.0)	0 (0)	0.116 †
**Hematological disorder**	1 (20.0)	2 (5.3)	0.316 †
TAM	1 (20.0)	2 (5.3)	0.316 †

ASD, atrial septal defect; AVSD, atrioventricular septal defect, DORV, double outlet right ventricle; DS, Down syndrome; FPIES, food protein-induced enterocolitis syndrome; PDA, patent ductus arteriosus; PVP, pulmonary valve prolapse; TAM, transient abnormal myelopoiesis; TOF, tetralogy of Fallot; VSD, ventricular septal defect. † Fisher’s exact test. ‡ Mann–Whitney U test.

**Table 2 nutrients-14-00388-t002:** Surgical history of children with DS.

	FPIES (*n* = 5)	Non–FPIES (*n* = 38)	*p*-Value †
**Surgical history (n (%))**	3 (60.0)	12 (31.6)	0.324
**Surgery for cardiac disease (n (%))**	1 (20.0)	10 (26.3)	1.000
Intracardiac repair	0 (0)	5 (13.2)	1.000
Ductus arteriosus ligation	0 (0)	3 (7.9)	1.000
BT shunt	0 (0)	2 (5.3)	1.000
PA banding	1 (20.0)	0 (0)	0.116
**Surgery for gastrointestinal disease (n (%))**	2 (40.0)	4 (10.5) §	0.136
Duodenal atresia repair	0 (0)	3 (7.9) §	1.000
Colostomy	2 (40.0)	1 (2.6)	0.032 *

BT, Blalock-Taussig; DS, Down syndrome; FPIES, food protein-induced enterocolitis syndrome; PA, pulmonary artery. † Fisher’s exact test. § Two of them underwent surgery for either intracardiac repair or BT shunt. * Significant at *p* < 0.05.

**Table 3 nutrients-14-00388-t003:** Serum total IgE level in children with DS.

	FPIES (*n* = 5)	Non–FPIES (*n* = 6)	*p*-Value
Male (n (%))	3 (60.0)	3 (50.0)	1.000 †
Median age at total IgE test (months (range))	3 (1–12)	10 (1–12)	0.230 ‡
Median total IgE (IU/mL (range))	<11 (<11–11)	<11 (<11–16)	1.000 ‡

DS, Down syndrome; FPIES, food protein-induced enterocolitis syndrome; IgE, immunoglobin E. † Fisher’s exact test. ‡ Mann–Whitney U test.

**Table 4 nutrients-14-00388-t004:** Clinical features of FPIES in children with DS.

Participant	1 [15]	2	3	4 [15]	5
**Sex**	male	male	male	female	female
**Age at onset (days)**	7	84	321	64	104
**Causative foods**	CM	CM	wheat	CM	CM
**Clinical symptoms**					
Vomiting	+	+	+	+	+
Diarrhea	+	+	+	+	+
Bloody stools	–	+	–	+	–
Abdominal distention	+	N/A	N/A	N/A	+
Fever	–	+	+	+	+
Metabolic acidosis	+	+	–	+	+
Dehydration/Shock	–	–	–	+	+
**Severity** [16]	severe	severe	severe	severe	severe
**Comorbidities**					
Cardiac disease	–	+ (TOF)	–	+ (ASD)	+ (AVSD)
Gastrointestinal disease	–	–	–	+ (rectovaginal fistula)	+ (imperforate anus)
Hematological disorder	–	–	+ (TAM)	–	–
**Surgical history**	N/A	PA banding	N/A	colostomy	colostomy
**Age at surgery (days)**	N/A	50	N/A	63	1
**Age at tolerance (months)**	132	26	N/A	49	18
**Total IgE (IU/mL)**	11	<11	<11	<11	<11
**Antigen-specific IgE (kU_A_/L)**	<0.35	<0.35	<0.35	<0.35	<0.35

ASD, atrial septal defect; AVSD, atrioventricular septal defect; CM, cow’s milk formula; DS, Down syndrome; FPIES, food protein-induced enterocolitis syndrome; IgE, immunoglobin E; N/A, not applicable; PA, pulmonary artery; TAM, transient abnormal myelopoiesis; TOF, tetralogy of Fallot.

## Data Availability

The data presented in this study are available on request from the corresponding author. The data are not publicly available due to privacy.

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
