# Peer review of "Food Protein-Induced Enterocolitis Syndrome in Children with Down Syndrome: A Pilot Case-Control Study"

_nutrients, 2022, doi:10.3390/nu14020388_

Round 1

Reviewer 1 Report

Dear Authors,

I find your article very interesting. The paper presents the medical history of 43 patients. The symptoms and medical procedures used for patients were analyzed. The conclusions that the authors drew are the same as for other allergic diseases "skip the causative foods". It is not spectacular, but it is still the best treatment. I do not see any problems with publishing the work in its current form. New scientific data would bring new ones to the work, but the most important thing in the treatment of human diseases is to share experience, especially if these are rare diseases.

Anyway, I'm just curious and will be happy with a few answers:

1) Why didn't moms decide to breastfeed their babies? I understand that hospitalization makes it difficult, but was it a trend or a requirement of experience?

2) Have you ever checked the profile of a fecal microbiota?

3) Have you checked the immune cell profile (e. g. T cell profile) in patients? You discuss this in the text (from line 194), but how is it possible that you had drops of blood for cytometric analysis of lymphocytes and cytokines?

Good luck:)

Reviewer 2 Report

In this study by Okazaki et al analyze the several clinical features of food protein-induced enterocolitis syndrome (FPIES) among the children with Down syndrome (DS). FPIES is a non-immunoglobulin E mediated gastrointestinal food hypersensitivity that manifests as profuse, repetitive vomiting, sometimes with diarrhea, leading to dehydration and lethargy with chronic metabolic derangements. This disease primarily affects infants. It is most commonly caused by cow's milk or soy protein, although other foods can be triggers. The authors described that surgical history of colostomy and followed nutritional formula containing cow milk can cause the onset of FPIES, suggesting amino acid based formula should be considered in the infants with such surgical history. In addition, the authors found low IgE levels in infants with DS suggesting non-IgE mediated hypersensitivity in such infant patients. Overall, this is well written manuscript and reports out significant findings which could definitely be interesting to the readers.

Indeed, FPIES is a non-IgE-mediated food hypersensitivity disorder with potential for severe dehydration and hypovolemic shock. However, further work is warranted to elucidate the immunologic basis of this disorder. A role of T cells in the local intestinal inflammation has been suggested similar to the previous findings, but it needs to be confirmed. The characteristics of the intestinal inflammatory response are largely determined by the cytokine release triggered by the pathologic mechanism. The lack of data on cytokine profile and metabolomics deficit our understanding of the disease. Humoral immune response may also be involved in the pathophysiology of FPIES with an increase of specific IgA and a decrease in specific IgG4 antibodies. The authors should discuss more on this in discussion section.  What do authors think about the possible immunological maternal strategies to ensure the optimal protection in DS children with FPIES? How do the clinical features and metabolic profile could differ from reported cases of adult FPIES if any or DS adult? The authors should further elaborate this on discussion section.
